# Prioritizing Solutions and Improving Resources among Young Pediatric Brain Tumor Survivors: Results of an Online Survey

Marco Bonanno [1,2,*], Claude Julie Bourque [3], Lye-Ann Robichaud [2,4], Ariane Levesque [2,4], Ariane Lacoste-Julien [2], Émélie Rondeau [2], Émilie Dubé [2], Michelle Leblanc [2], Marie-Claude Bertrand [1], Carole Provost [1], Leandra Desjardins [2,5] and Serge Sultan [2,4]

1  Hematology-Oncology Department, Sainte-Justine University Hospital Center, Montreal, QC H3T 1C5, Canada
2  Psycho-Oncology Center (CPO), Sainte-Justine University Hospital Research Center, Montreal, QC H3T 1C5, Canada
3  Department of Pediatrics, Université de Montréal, Montreal, QC H3T 1C5, Canada; claude.julie.bourque@umontreal.ca
4  Department of Psychology, Université de Montréal, Montreal, QC H2V 2S9, Canada
5  CHU Sainte-Justine Research Centre, Montreal, QC H3T 1C5, Canada
*  Correspondence: marco.bonanno.hsj@ssss.gouv.qc.ca

**Abstract:** Pediatric Brain Tumor Survivors (PBTS) often experience social, academic and employment difficulties during aftercare. Despite their needs, they often do not use the services available to them. Following a previous qualitative study, we formulated solutions to help support PBTS return to daily activities after treatment completion. The present study aims to confirm and prioritize these solutions with a larger sample. We used a mixed-methods survey with 68 participants (43 survivors, 25 parents, PBTS' age: 15–39 years). Firstly, we collected information about health condition, and school/work experience in aftercare. Then, we asked participants to prioritize the previously identified solutions using Likert scales and open-ended questions. We used descriptive and inferential statistics to analyze data, and qualitative information to support participants' responses. Participants prioritized the need for evaluation, counseling, and follow-up by health professionals to better understand their post-treatment needs, obtain help to access adapted services, and receive information about resources at school/work. Responses to open-ended questions highlighted major challenges regarding the implementation of professionals' recommendations at school/work and the need for timely interventions. These results will help refine solutions for PBTS and provide key elements for future implementation. Translating these priorities into action will need further work involving professionals and decision makers.

**Keywords:** adolescent/young adult; brain tumor; parents; survivorship; survey

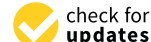



## 1. Introduction

Brain tumors are the second most common group of pediatric cancers, after blood and lymphoma cancers, and the most common in children under 15 years. Due to improvements in detection and intervention, the five-year survival rate of this population has increased over time and now exceeds 80% in most high-income countries [1–3]. Adolescent and young adult pediatric brain tumor survivors (PBTS) aged 15 to 39 years have been identified as a distinct and more vulnerable subpopulation of patients compared to survivors of other types of cancer [4,5]. Because of the disease and the multiple intense treatments involving transcranial radiation, surgery, and chemotherapy, most PBTS are at high risk of developing major chronic health issues that impact short and long-term social integration, academic performance and employment [6,7]. Notably, the specific neurocognitive, psychosocial and physical difficulties they experience may cause important disruptions to the transition from the pediatric to the adult healthcare system and hamper the return to daily activities.

This occurs at a time when young people are particularly vulnerable and face daunting developmental challenges (finding a job, building romantic relationships, being a parent, etc.) [4,8–10].

In this context, it is no surprise that young brain tumor survivors report the poorest health-related quality of life among all childhood cancer survivors and have specific needs [11–13]. For instance, it is estimated that the majority of PBTS show impairments in at least one neurocognitive domain in adulthood [14,15]. Despite their needs, pediatric brain tumors have been less studied than other forms of cancer (e.g., leukemia) due to the heterogeneity of clinical history and their complex neuro-psychosocial late effects [16]. Available support programs and services have often been designed to reach a broader population of cancer survivors and have not been developed or validated for this specific population. It is a widespread observation that current available services are underutilized by PBTS, as they do not adequately meet their specific needs or take into account their functional limitations [10,17].

A better understanding of the challenges experienced by PBTS in the post-treatment period and improved tailored services could help patients to better cope with late effects [6]. Rare initiatives have demonstrated that early adapted interventions specifically addressed to this population could be beneficial to improve quality of life and facilitate re-entry into daily life activities, school and work [18,19]. These interventions have focused on psychological interventions, including strategies related to distress management, providing information about accessible resources, and counseling to improve school and employment integration. Yet, little is known about evidence-based interventions specifically targeting this population, which can make it difficult to plan concrete and relevant programs that aim to meet the needs and challenges of PBTS [20]. There is a need for more empirical data and users' guidance to adapt existing resources, develop new resources, and implement them into practice [10,21,22].

This study is the second step of a larger research project of which the primary goal aims to improve available resources and formulate future interventions to meet the specific needs of PBTS during aftercare. Several steps are needed to achieve this goal in order to identify, confirm and operationalize solutions in clinical practice. The first step was recently completed and consisted in generating ideas of solutions based on survivors' expressed needs [18]. We used a qualitative methodology with three sequential focus groups with survivors, parents and clinicians (*n* = 22). This allowed for 14 different solutions to emerge in the areas of: returning to daily life activities (e.g., evaluation and ongoing follow-ups by health professionals, liaison from hospital to daily life sectors), support needs (e.g., study and work counseling), and information needs (e.g., adapted services and available resources) [18]. In the present study, we aimed to validate and prioritize these ideas in a larger sample.

Our general objective was to explore, in a larger sample of users, the relevance and usefulness of solutions to gaps in supportive aftercare services formulated in the previous qualitative study [18]. Our specific objectives were to (1) yield a portrait of perceived difficulties of survivors during aftercare in the health, education and employment domains, and (2) prioritize the previously identified solutions, as well as collect further information and ideas to translate these solutions into practice.

## 2. Materials and Methods

### 2.1. Design

The methodological approach of this study is based on a convergent mixed design [23,24]. This allows to explore a research problem using different methods to complete and deepen the information collected. Our strategy aimed to (1) collect quantitative data through an online survey from a large sample to statistically describe the relevance of candidate ideas from the previous qualitative study, and (2) enrich and support responses of participants with contextual qualitative information.

### 2.2. Participants and Recruitment

Participants were adolescent and young adult pediatric brain tumor survivors (AYA-PBTS) and AYA-PBTS' parents. Inclusion criteria were: (1) survivor aged 15 to 39 years; (2) diagnosed between 0 and 18 years; (3) active cancer-focused treatment completed for more than 12 months (surgery, chemotherapy, radiotherapy, proton therapy, immunotherapy, cell or gene therapy); (4) able to use a computer, tablet or smartphone, or other digital tools to answer an online questionnaire; (5) able to read and write in English or French. There were no exclusion criteria. Parents had to respond to the same criteria for their child.

Participants were recruited from the long-term follow-up clinic (LTFU) at the Sainte-Justine University Health Centre oncology clinic and at two community supportive organizations working in the oncology domain (Leucan and Brain Tumour Foundation of Canada). Potential participants were identified from existing databases by nurses from the LTFU and by managers from these organizations, who sent potential participants an email explaining the research. Participants who were willing to participate in this study were sent a hyperlink by a member of the research team. This hyperlink allowed participants to connect to an online questionnaire that included a consent form. We also asked professional organizations in Canada to advertise the research on social media, directing potential participants to the survey platform. Hence, sampling was convenient and non-probabilistic. All respondents gave their informed consent electronically.

### 2.3. Data Collection

The survey was pretested by expert patients and members of the team to take approximately 35 min to complete. The pretesting also allowed the survey to be adapted to the possible cognitive and attentional limitations of the users. The survey was co-designed with two young survivors (A.L.-J. and É.D.), one parent (M.L.), and one neuropsychologist of our unit (M.-C.B.). We collected data through sociodemographic questions, multiple-choice and open-ended questions, Likert scales, and respondents could add free comments (the full survey is available in Supplementary Table S1).

The survey was composed of two sections according to the specific aims of the study. The first part was made up of questions designed to describe the experience and perceived difficulties of survivors during aftercare in the domains of health, education and employment. The second part was designed to inform and evaluate the relevance of 14 solutions aiming to meet the needs at re-entry during aftercare. Participants were asked to share their opinions on the relevance of these solutions and rate their usefulness. Solutions dealt with: (1) transition services between cancer-focused care and return to daily activities, (2) practical information (e.g., stores selling adapted material), information on specific needs related to long-term sequelae and about the disease, support and guidance for social, academic and professional reintegration (between peers and with stakeholders). For parent respondents, the wording was adapted to allow them to describe their child's status. We used the Survey Monkey® platform (SurveyMonkey Inc., San Mateo, California, USA) to administer the survey.

### 2.4. Data Analysis

Statistical analyses were performed with IBM SPSS Statistics version 27.0 (IBM Corp., Armonk, NY, USA) and an alpha level of 0.05 was set for statistical significance. All text answers to the questions were collected in a document file, according to informant and domain, to support and deepen the quantitative data, using a descriptive thematic analysis approach [25,26]. Analyses were performed both for the total sample and by respondent group (young survivors and parents). Only completed surveys were used in the data analysis.

Analyses were performed according to the specific aims of the study using descriptive and inferential statistics. First, we used descriptive statistics (percentage, means, medians and standard deviation) to calculate respondents' sociodemographic information, clinical history, school and employment attendance. Then, we performed inferential statistics

to analyze the experience and difficulties of survivors during aftercare in the domains of health, school and employment. We used a Wilcoxon test within the total sample to compare health status difficulties and perceptions. A Mann–Whitney test was performed to compare ranks between the responses of both informant populations. For the evaluation of perceived difficulties at school, we performed a Kruskall–Wallis and a Mann–Whitney Test to compare responses and ranks. A Chi-square test was used to compare higher difficulties in the total sample in the domain of employment. Finally, to determine which solutions received the highest endorsements in respondents' responses, we compared means using repeated ANOVA and a Bonferroni post hoc test.

## 3. Results

### 3.1. Participant Characteristics

A total of 109 potential participants logged on to the survey. Of these, 41 (38%) were discarded as they did not meet the inclusion criteria (i.e., responded "no" to the first item, $n = 26$), or stopped after responding to less than the first five sociodemographic items ($n = 15$). Thus, we analyzed the responses of the 68 remaining respondents. Among them, 43 (63%) were pediatric brain tumor survivors (PBTS) aged 15–39 years (M = 24 ± 6) and 25 (37%) were parents aged 43–62 years (M = 51 ± 6 years, child 16–31 years M = 22 ± 5 years). A sociodemographic and clinical description of the sample is available in Table 1. Overall, survivors had been treated for a medulloblastoma (40%), astrocytoma (33%) or other forms of brain tumors (27%). At time of the survey, they reported being followed in a pediatric (36%) or adult (54%) hospital setting, or in rare cases by a family doctor (3%). Only seven percent reported having no follow-up. Respondents lived in Canada, with survivors (74%) and parents (91%) mainly from the province of Quebec (Table 1). In the present report, we considered survivors and parents as independent samples (see limitations).

**Table 1.** Sociodemographic description and clinical history of survey participants.

| Characteristics | | Survivors (Self-Report) (N = 43) | | Survivors (Parent-Report) (N = 25) | | Total (N = 68) | |
|---|---|---|---|---|---|---|---|
| | | N (%) | Mean ± SD | N (%) | Mean ± SD | N (%) | Mean ± SD |
| Language | French | 29 (67.4) | | 22 (88) | | 51 (75) | |
| | English | 14 (32.6) | | 3 (12) | | 17 (25) | |
| Sex | Male | 11 (25.6) | | 19 (76) | | 30 (44.1) | |
| | Female | 32 (74.4) | | 6 (24) | | 38 (55.9) | |
| Age | At survey | | 25.09 ± 6.15 | | 21.68 ± 4.87 | | 23.84 ± 5.91 |
| | At diagnosis | | 11.81 ± 6.61 | | 10.68 ± 5.23 | | 11.4 ± 6.12 |
| | At the end of treatments | | 15.19 ± 7.49 | | 13.33 ± 5.46 | | 14.52 ± 6.84 |
| Family situation | Single | 27 (65.9) | | 18 (75) | | 45 (69.2) | |
| | Separated/divorced | 1 (2.4) | | - | | 1 (1.5) | |
| | With children | 4 (9.8) | | 1 (4.2) | | 5 (7.7) | |
| | Missing | 2 (4.65) | | 1 (4) | | 3 (4.41) | |
| Place of living (age ≥ 20 years) ** | With parent(s) | 18 (56.3) | | 10 (71.4) | | 28 (60.9) | |
| | Alone | 4 (12.5) | | 2 (14.3) | | 6 (13) | |
| | With roommate | 1 (3.1) | | - | | 1 (2.2) | |
| | With partner | 9 (28.1) | | 2 (14.3) | | 11 (23.9) | |
| | Missing | 2 (5.9) | | 1 (6.7) | | 3 (6.3) | |
| Diagnosis | Medulloblastoma | 16 (41) | | 10 (40) | | 26 (40.6) | |
| | Germ cell tumor | 3 (7.7) | | 2 (8) | | 5 (7.8) | |
| | Ependynoma | 2 (5.1) | | 2 (8) | | 4 (6.3) | |
| | Astrocytoma * | 14 (35.9) | | 7 (28) | | 21 (32.8) | |
| | Cranio-pharyngioma | 1 (2.6) | | 3 (12) | | 4 (6.3) | |
| | Other | 3 (7.7) | | 1 (4) | | 4 (6.3) | |
| | Do not remember | 4 (9.3) | | - | | 4 (5.9) | |
| Treatments | Radiotherapy | 32 (74) | | 23 (95.8) | | 55 (82.1) | |
| | Chemotherapy | 30 (69.8) | | 19 (79.2) | | 49 (73.1) | |
| | Surgery | 35 (81.4) | | 21 (87.5) | | 56 (83.6) | |

**Table 1.** *Cont.*

| Characteristics | | Survivors (Self-Report) (N = 43) | | Survivors (Parent-Report) (N = 25) | | Total (N = 68) | |
|---|---|---|---|---|---|---|---|
| | | N (%) | Mean ± SD | N (%) | Mean ± SD | N (%) | Mean ± SD |
| Medical Follow-up | Pediatric hosp. | 13 (30.2) | | 11 (45.8) | | 24 (35.8) | |
| | Adult hosp. | 25 (58.1) | | 11 (45.8) | | 36 (53.7) | |
| | Family doctor | 2 (4.7) | | - | | 2 (3) | |
| | No follow-up | 3 (7) | | 2 (8.3) | | 5 (7.5) | |
| | Missing | - | | 1 (4) | | (1.47) | |

\* Glioblastoma, optic pathway glioma, pilocytic astrocytoma. \*\* The 20-year cut-point was considered to help compare frequencies with the official census from Statistics Canada [27].

### 3.2. The Experience and Difficulties of Survivors during Aftercare in the Domains of Health, School, and Employment

### 3.2.1. Health Domain

Of the survey respondents, 72% (*n* = 48) reported that their health status was worse or much worse than peers and 85% (*n* = 57) reported difficulties regarding their health (Supplementary Table S2). Among the difficulties reported by the total sample, physical issues were rated as having the most impact on daily life in comparison to psychological (Z = −2.952 *p*= 0.010) and social issues (Z = −2.396 *p* = 0.017). Similarly, cognitive issues were rated as more impactful than psychological (Z = −3.121 *p* = 0.002) and social (Z = −2.322 *p* = 0.020) difficulties (Figure 1). No significant difference between survivors' and parents' ratings were identified (*p* values ≥ 0.407).

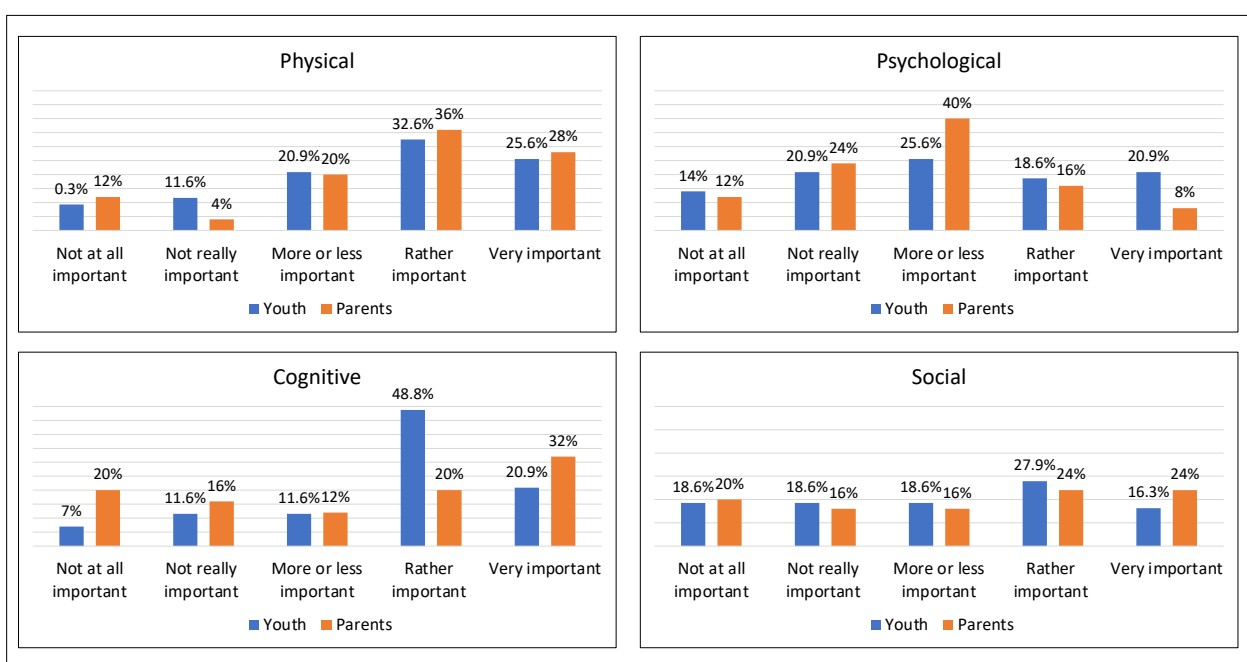

**Figure 1.** Health status—Comparison of reported difficulties in four domains across informant (brain tumor survivors or parents) in 68 survey respondents. Note: Item wording was: What is the impact of these difficulties on daily life?

### 3.2.2. School Domain

At the time of survey completion, 52% (*n* = 34) of survivors were attending school. Those attending school were aged 16–34 years (M = 20 ± 4 years). Among survivors aged ≤ 18 years, all were attending school, whereas this proportion was 57% for those aged 19–24 years, and 23% for those aged 25–34 years. Within these two groups, 29% (*n* = 10) studied at the university level (19–24 years group = 18%, *n* = 5; 25–34 years group 83%, *n* = 5). For those who were not attending school at time of survey (mean age: 27.38 ± 5.4, range: 19–39), the levels of education reached were high school (28%,

aged M = 25 ± 4), pre-university, adult post high-school diploma or professional post high-school diploma (56%, aged M = 27 ± 4), university bachelor's or master's degree (13%, aged M = 36.5 ± 6).

When reporting retrospectively on schooling during cancer treatments, only 24% (*n* = 16) of survivors had kept going to school, whereas 43% (*n* = 29) had completely stopped (time out of school = 16 ± 11 months, range 2–60 months), and 33% (*n* = 22) had been helped by a teacher at home (mean follow-up =12 ± 8 months, range 1–36 months). When describing the difficulties in school attendance during aftercare (Figure 2), we found differences among these three groups ($\chi^2(2)$ = 8.459, *p* = 0.015). Perceived difficulties were rated higher when survivors had taken a full break from school during cancer treatments (Figure 2 Panel B) compared with those who had kept going to school (U = 120.500, *p* = 0.009) and those who had been helped by a teacher at home (Figure 2 Panel C, U = 216.000, *p* = 0.036). We found no differences between those who had kept going to school and those who had been helped at home (U = 126.500, *p* = 0.186). No significant difference between survivors' and parents' ratings were identified.

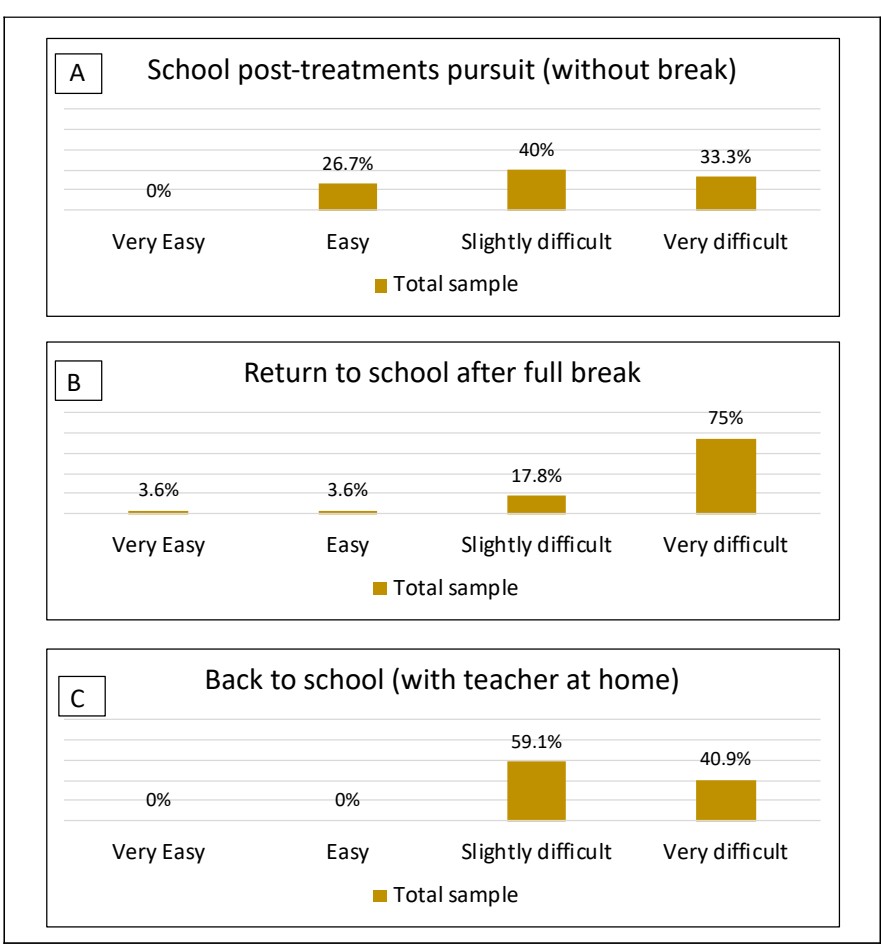

**Figure 2.** Perceived difficulties of school attendance in aftercare according to schooling during treatment in survey respondents (*n* = 65). Note: Item wording was: How would you describe your experience of continuing your studies [of returning to school] after the end of your treatments?

Respondents had the opportunity to freely point out what they considered helpful and difficult in their school experience (or that of their child) during aftercare. The questions asked were What helped you the most in your return [the return of your child] at school? And What was the hardest part of going back at school for you [for your child]? Factors mentioned as helpful were parental guidance, having good friends, being supported by teachers and school principals regarding their post-treatment needs, and having medical

follow-ups adapted to the school schedule. In this regard, a young survivor said: "My parents never let me down in my studies...the teachers and school principal were understanding". When listing issues, participants pointed out a variety of physical and cognitive difficulties that negatively impacted their performance, such as fatigue, concentration, learning and language problems, organizational problems, visual problems, and motor skills problems. A young survivor, in this regard, said: "It was difficult to manage the changes in my cognitive abilities such as my ability to multitask or to concentrate. I also had a lot of difficulties with extreme fatigue and I was unable to tolerate loud noises and busy environments". Others described their difficulties when re-entering school, and especially highlighted their difficulties with peer relationships. They described difficulties in managing physical differences with others, being bullied, losing friends, or having difficulty to make new friends. Respondents also mentioned the teachers' lack of sensitivity towards their important needs. A young survivor summarized his status as follows: "Everything was difficult...studying, being bullied, being a different child, attending school, teachers not understanding that I needed more time and help...".

### 3.2.3. Employment Domain

In line with the Statistics Canada census age categories, we found that among survivors of 15–24 years ($n = 37$) 27% ($n = 10$) had never worked. This proportion was of 3% ($n = 1$) among the 25+ years ($n = 29$). Among the younger group, 38% ($n = 14$) had already worked but were no longer working or 35% ($n = 13$) were working at the time of the survey. Among the older group, 31% ($n = 9$) had already worked but were no longer working or 66% ($n = 19$) were working at the time of the survey. For the younger group (15–24 years) who had some work experience, we observed that 52% ($n = 14$) were employed for less than one year, 44% ($n = 12$) between one to five years, and 4% ($n = 1$) more than five years. Within this group, 19% ($n = 5$) reported working part-time for less than 7 h/week, 70% ($n = 19$) for 7–30 h/week, and 11% ($n = 3$) reported working more than 30 h/week. For the older group (25+ years) who had some work experience, we noted that 31% ($n = 9$) had been employed for less than one year, 54% ($n = 14$) between one to five years, and 15% ($n = 4$) for more than five years. Within this group, 11% ($n = 3$) reported working part-time for less than 7 h/week, 44.5% ($n = 12$) for 7–30 h/week, and 44.5% ($n = 12$) reported working more than 30 h/week.

Respondents were asked to describe their experience in searching for a job and their ability to carry out tasks at work. Among them, 57% ($n = 30$) found it slightly–very difficult to find a job and 56% ($n = 29$) found it slightly–very difficult to carry out tasks at work. Notably, parents and survivors rated these items the same way ($p$ values > 0.440). When given the opportunity to comment qualitatively, participants gave some examples of what they found helpful in the field of employment. They mentioned topics such as the support received from parents, friends, universities and other organizations to find a job. They insisted on the interest of developing personal strategies to better perform at work, having more flexibility in work conditions, and holding a position more adapted to their health status and difficulties. A young survivor commented on facilitating factors at work, as follows: "Writing notes, making eye contact with customers, asking them to repeat if I cannot hear them, asking the other employees for assistance when I have a question". On the opposite, respondents mentioned some factors that made working experiences more difficult. They pointed out the complexity of administrative procedures to look for a job and the cognitive challenges they met when confronted with the need to multitask. They also highlighted their memory and concentration problems, the stress and fatigue that limited their daily work tasks, and the particularly intense challenges experienced when communicating with others, such as participating in meetings with several people because of the hearing and concentration problems they experienced. A parent explained the difficulties encountered by her child at work as follows: "She needs to be coached to concretely learn how to do the tasks, she has a slow execution so she needs more time to learn, she does not have a lot of physical strength, contact and communication

with clients/work colleagues/employers is difficult because of the lack of social skills and understanding".

### 3.3. Validating and Prioritizing Solutions

Here, we suggested a series of solutions to deal with the identified challenges of aftercare and early survivorship. Solutions were derived from a previous qualitative study [18]. Each solution was rated by respondents on a scale from 0 (not at all helpful) to 3 (very helpful). As shown in Figure 3, all solutions were judged as helpful on average (Median mean = 2.42/3). A repeated measures ANOVA with a Greenhouse–Geisser correction determined that ratings differed significantly among solutions specified in Table 2 (F (8.401) = 5.459, $p < 0.001$). Post hoc comparisons showed that Solution #7 appeared significantly less helpful than #1, #2, #3, #5, #12, and #13. Solution #10 also appeared less helpful than #5. Other comparisons were not significant (full results in supplementary Table S3). Five solutions obtained the most interest among participants and, notably, were rated highly by both survivors and parents. These solutions related to the topics of regular follow-ups for support (#5, M = 2.70, SD ± 0.63, CV = 23.7%), information on school resources (#13, M = 2.69, SD ± 0.76, CV = 28.2%), evaluation (#1, M = 2.68 ± SD± 0.56, CV = 21.1%), liaison between hospital and school/work (#3, M = 2.63, SD ± 0.75, CV = 28.6%), and counseling (#2, M = 2.62, SD ± 0.7, CV = 26.9%). Receiving advice on relationships and sexuality was judged as the least helpful (#7, M = 1.97, SD =1.02, CV = 52%). When exploring scores across informants, survivors' endorsements of solutions tended to be rated lower than parents' (Median mean= 2.33/3 vs. 2.61/3). The difference was statistically significant for Solutions #1, #2, #3, and #9.

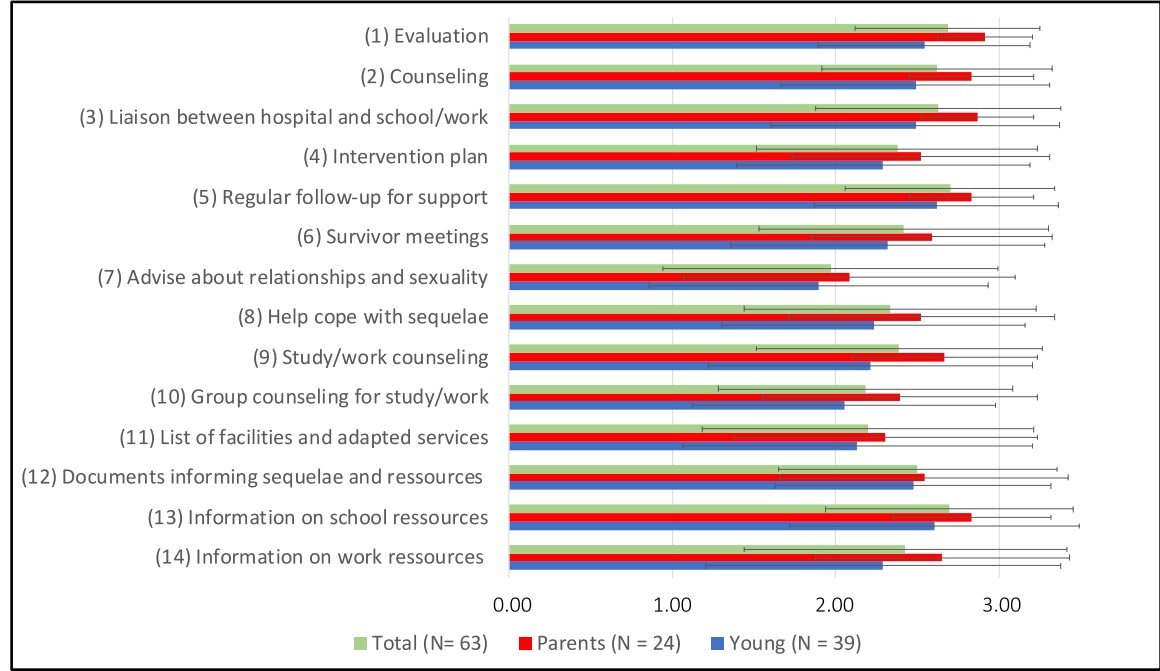

**Figure 3.** Mean and standard deviation of relevance judgments for each solution. Note: Values for the responses to solutions: 0 (Not at all helpful), 1 (Slightly helpful), 2 (Pretty helpful), 3 (Very helpful).

Table 2. Assessment of whether suggested solutions are considered helpful by PBTS survivors and parents (*n* = 68).

| Solutions | Helpful? | Young Survivors N (%) | Parents N (%) | Total N (%) |
|---|---|---|---|---|
| (1) When the treatments are finished, be evaluated by hospital professionals to better understand your needs for the return to your daily activities | Not at all<br>Slightly<br>Pretty<br>Very | -<br>3 (8.1)<br>11 (29.7)<br>23 (62.2) | -<br>-<br>2 (8.7)<br>21 (91.3) | -<br>3 (5.0)<br>13 (21.7)<br>44 (73.3) |
| (2) Receive advice and recommendations from hospital professionals to help you explain your situation and request appropriate services at school or at work | Not at all<br>Slightly<br>Pretty<br>Very | 1 (2.6)<br>5 (12.8)<br>7 (17.9)<br>26 (66.7) | -<br>-<br>4 (16.7)<br>20 (83.3) | 1(1.6)<br>5 (7.9)<br>11 (7.5)<br>46 (73) |
| (3) Have a hospital professional contact the school or workplace and offer recommendations for services tailored to your needs | Not at all<br>Slightly<br>Pretty<br>Very | 3 (7.7)<br>1 (2.6)<br>9 (23.1)<br>26 (66.7) | -<br>-<br>3 (13)<br>20 (87) | 3 (4.8)<br>1 (1.6)<br>12 (19.4)<br>46 (74.2) |
| (4) Participate in a meeting with members of the family, professionals and workers from school/work to discuss needs and organize services for the reentry | Not at all<br>Slightly<br>Pretty<br>Very | 3 (7.9)<br>2 (5.3)<br>14 (36.8)<br>19 (50) | -<br>4 (17.4)<br>3 (13.0)<br>16 (69.6) | 3 (4.9)<br>6 (9.8)<br>17 (27.9)<br>35 (57.4) |
| (5) After resuming your daily activities, have access to regular follow-ups by hospital professionals, if needed, to receive support or evaluations | Not at all<br>Slightly<br>Pretty<br>Very | 2 (5.1)<br>-<br>9 (23.1)<br>28 (71.8) | -<br>-<br>4 (16.7)<br>20 (83.3) | 2 (3.2)<br>-<br>13 (20.6)<br>48 (76.2) |
| (6) Meet other young people who are going through a similar situation to talk. This could be done over the internet on a secure site or during an activity | Not at all<br>Slightly<br>Pretty<br>Very | 2 (5.3)<br>7 (18.4)<br>6 (15.8)<br>23 (60.5) | 1 (4.5)<br>-<br>6 (27.3)<br>15 (68.2) | 3 (5.0)<br>7 (11.7)<br>12 (20)<br>38 (63.3) |
| (7) Meet with a hospital professional who can answer questions about friendships, romantic relationships, or sexuality | Not at all<br>Slightly<br>Pretty<br>Very | 5 (13.2)<br>7 (18.4)<br>13 (34.2)<br>13 (34.2) | 2 (8.3)<br>5 (20.8)<br>6 (25)<br>11 (45.8) | 7 (11.3)<br>12 (19.4)<br>19 (30.6)<br>24 (38.7) |
| (8) Being able to count on the support of a professional from the hospital or another former patient to help cope with the difficulties of daily life following cancer | Not at all<br>Slightly<br>Pretty<br>Very | 2 (5.1)<br>7 (17.9)<br>10 (25.6)<br>20 (51.3) | -<br>4 (19)<br>2 (9.5)<br>15 (71.4) | 2 (3.3)<br>11 (18.3)<br>12 (20)<br>35 (58.3) |
| (9) Receive advice from a professional who helps choose an area of study or employment, taking into account strengths, limitations and interests, and who gives advice at different stages of studies/employment | Not at all<br>Slightly<br>Pretty<br>Very | 3 (7.9)<br>6 (15.8)<br>9 (23.7)<br>20 (52.6) | -<br>1 (4.2)<br>6 (25)<br>17 (70.8) | 3 (4.8)<br>7 (11.3)<br>15 (24.2)<br>37 (59.7) |
| (10) Participate in school or professional orientation group meetings to discuss studies and work with other young people who are living a similar situation | Not at all<br>Slightly<br>Pretty<br>Very | 3 (7)<br>6 (14)<br>15 (34.9)<br>14 (32.6) | 1 (4)<br>2 (8)<br>7 (28)<br>13 (52) | 4(6.6)<br>8 (13.1)<br>22 (36.1)<br>27 (44.3) |
| (11) Have a list of places and services where you can find suitable equipment for everyday life to meet your difficulties | Not at all<br>Slightly<br>Pretty<br>Very | 5 (11.6)<br>4 (9.3)<br>10 (23.3)<br>19 (44.2) | 2 (8)<br>1 (4)<br>8 (32)<br>12 (48) | 7 (11.5)<br>5 (8.2)<br>18 (29.5)<br>31 (50.8) |
| (12) Have documents that describe any difficulties that might be encountered after finishing treatments and the best solutions and resources available to help overcome them | Not at all<br>Slightly<br>Pretty<br>Very | 2 (5.6)<br>2 (5.6)<br>9 (25)<br>23 (63.9) | 2 (8.3)<br>-<br>5 (20.8)<br>17 (70.8) | 4 (6.7)<br>2 (3.3)<br>14 (23.3)<br>40 (66.7) |
| (13) Have information on resources and solutions available to help you in school | Not at all<br>Slightly<br>Pretty<br>Very | 3 (7.9)<br>1 (2.6)<br>4 (10.5)<br>30 (78.9) | -<br>1 (4.2)<br>2 (8.3)<br>21 (87.5) | 3 (4.8)<br>2 (3.2)<br>6 (9.7)<br>51 (82.3) |
| (14) Have information on the resources and solutions available to help at work | Not at all<br>Slightly<br>Pretty<br>Very | 6 (15.8)<br>-<br>9 (23.7)<br>23 (60.5) | 1 (4.3)<br>1 (4.3)<br>3 (13)<br>18 (78.3) | 7 (11.5)<br>1 (1.6)<br>12 (19.7)<br>41 (67.2) |

When given the opportunity to comment, respondents gave their opinions and ideas to implement solutions and translate them into concrete actions. Here are some excerpts from young survivors and parents regarding the solutions with the most helpful and least helpful quantitative ratings (Figure 3).

Solution #1 regarded being evaluated by professionals to help survivors better understand their needs in aftercare. Respondents pointed out the need to receive evaluations at the right time, with involvement of the school and the possibility that this assessment

would take place outside the healthcare system. A parent wrote: "Side effects from the treatments appeared several years later. In some cases, I believe that the proposed evaluations should not be done systematically only at the end of the treatments". Solution #2 regarded receiving advice and recommendations from professionals to help survivors explain their healthcare situation and ask for appropriate services at school or at work. For this solution, comments converged on the importance of increasing the awareness of the non-medical environment to the "invisible" sequelae of survivors and to find ways to better transfer and apply recommendations when the environment changes. In this regard, a young survivor said: "I got this [solution] at school but a lot of teachers did not follow along because "I looked" fine. Huge struggle and it is the reason I can't do a normal course in college". Solution #3 suggested having a hospital professional liaising with the school or workplace and offer recommendations for services tailored to the needs of the young survivor. Respondents who reported having obtained this service found it helpful. A few stressed the lack of appropriate resources in school and the workplace to implement this solution, and the difficulty of receiving this help from healthcare professionals once the survivor is no longer a patient in a pediatric hospital. Supporting this solution, a parent said: "Yes, we had to go through alone and try to convince the school principals. Not very pleasant". Solution #5 proposed having access to regular follow-ups by hospital professionals after the survivor has resumed daily activities. The majority of respondents agreed with this solution due to the ongoing and specific needs of brain tumor population in aftercare. Some mentioned the challenge of obtaining services when young people get older, after 18 years. In this regard, a young survivor commented as follows: "I had this as a kid, but when I hit 18 and needed this, I did not have access to therapy without paying out of pocket". Solution #13 was about having information on resources and solutions available to help at school. Respondents pointed out the procedural steps required to obtain resources and services and suggested that this information should regularly be given at the end of treatments. A parent reported her experience: "She had this, but we had to fight to get it. Caused much unnecessary stress with an already difficult transition".

Solution #7 was about receiving counseling on friendships, romantic relationships, or sexuality. This was significantly less endorsed than the five previous solutions. For this solution, respondents had varying views on its utility. Some mentioned feeling uncomfortable and evaluated it as unnecessary, especially for the topic of sexuality. A young survivor summarized as follows: "Personally, I would be embarrassed to ask for help at this level, but I think that it could be useful for someone who has been socially isolated for a long time". In contrast, others explained that this could help regain self-confidence and improve relationships: "It would make us want to come out of our shell, and find a social life that will make us grow".

## 4. Discussion

This online survey is the second step of a larger study which aims to provide and improve services among adolescent and young adult survivors of a pediatric brain tumor. Our results allowed to provide more details on some of the difficulties faced by this vulnerable population in the psychosocial, cognitive and physical domains [4]. While different issues and sequelae have already been described in the PBTS population, little is known about their association with their sociodemographic context, such as their educational and occupational status, relationship status, independent living situation and disability [28]. Yet, studies investigating the sociodemographic context in PBTS population seem to be an important avenue to explore in order to better understand the extent of their difficulties compared to the general population and to adapt services accordingly.

Among respondents, 65% of those aged 20–34 years were living with their parents at the time of the survey. The rate for the same range is around 35% in the general population in Canada [27]. As for marital status, 64% of respondents aged 20–40 years declared being single, while the norm in Canada is 47% for the same age range [29]. Regarding school, 45% of young adults aged from 25 to 34 years in our survey completed post-secondary

education, compared to 73% in the general population [30]. For the employment domain, we found that 47% of our respondents aged 15 to 39 years worked compared to 74% in the general population [31]. A similar lower percentage among PBTS in social, educational and occupational status compared to the general population was described in other countries [28,32]. These results can be explained by the high percentage of impairments shown by this population in at least one domain of their life, which places PBTS in the one in five (22%) Canadians aged 15 years and over who have at least one disability [33–35].

It is particularly informative to compare employment rates of the present sample with the ones of people with a disability. In the population aged 15–34 years, the rate of employment is 54% for people with a mild disability and 41% for those with a moderate disability. This would place the present PBTS sample close to the employment rate of people with a moderate disability [36]. As for the severity of disability in the general population, PBTS show heterogenous and diverse difficulties [16]. According to the World Health Organization (WHO), the brain tumor population is classified in 4 grades, divided in low-grade (I–II) and high-grade (III–IV). Studies have shown that long-term health in low-grade illnesses is similar to that of the general population, while high-grade illnesses typically generate significantly more health difficulties [28]. These data highlight the importance of providing assessments, follow-ups and services to survivors, in accordance with the severity of their condition and their late effects throughout the different stages of life.

Sequelae experienced by PBTS, such as fatigue, can also affect the number of hours that they can dedicate to working. For instance, in the Canadian population with at least one incapacity, 37% of those aged 15–24 years and 58% of those aged 25–44 years work more than 30 h [37]. In comparison, the rates were 11% and 44.5% in the present sample, for comparable age groups. It is known that the job market lacks flexibility in adapting work to the needs of people with disabilities [38]. This may explain why 35% of our sample had stopped working at the time of survey. This supports the observation that PBTS are also the group with the highest unemployment rates among cancer survivors [39–41]. It will probably take a major effort at many levels to create better conditions to ease (re)entry in the workplace for a population as vulnerable as the one who participated in the present survey. In this line, authors have advocated in favor of regular targeted screening on employment issues experienced by PBTS [42]. It is also important to stress that optimal social reintegration should be encouraged by societies at all levels and in the long term, as demonstrated by the movement supporting the "oncological oblivion" or the "right to be forgotten", which aims to provide a legal framework to protects cancer survivors from financial discrimination [43]. This is also supported by recent studies that have begun to assess the interaction between medical/psychosocial health and socioeconomic hardship in cancer survivors. These studies have shown that financial discrimination can cause or potentially exacerbate physical and psychological harm in this population [44,45].

In this context, the solutions that obtained the most interest among our respondents were about receiving assessments, counselling and regular follow-ups, as well as providing liaison among different sectors, and information on available resources. Comments reported by participants on these solutions stated that services in aftercare were often neither equally nor timely distributed among survivors. Moreover, these services, if ever offered, were difficult to access compared to those during the treatment period. There is growing evidence, however, that PBTS need continuous support in aftercare [46]. In this regard, participants mentioned an important lack of continuity when survivors must change environments, transitioning from pediatric to adult healthcare, or in their reentry to school or work. Particularly, the return to school after a full break during cancer treatments seemed to be a major challenge in our sample. Finally, participants pointed out the lack of awareness among providers outside the healthcare sector regarding their difficulties and sequelae (Figure 4).

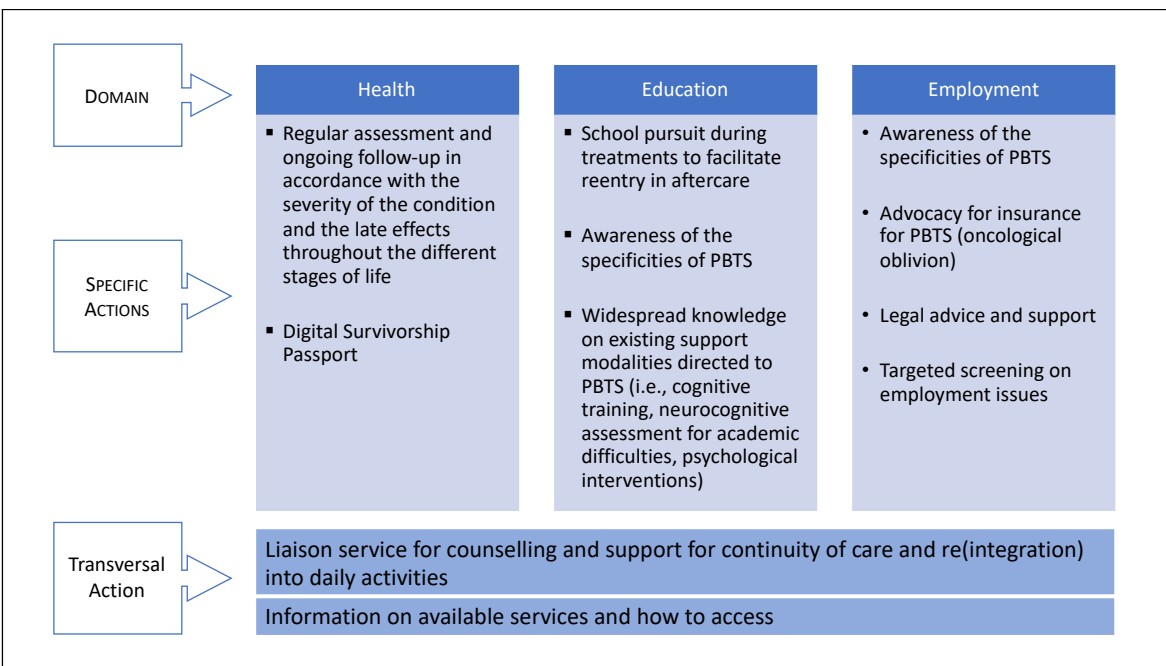

**Figure 4.** Suggested key actions to support social integration of pediatric brain tumor survivors.

The need for available services in aftercare is an important issue given the increasing number and aging of this vulnerable population of survivors [47]. We now know that as a result of cancer and its treatment, physical and cognitive sequelae often increase over time [3]. The dynamics of these late-effects challenges the maintenance of key developmental tasks or roles, such as having and keeping employment or developing romantic relationships [4,48,49]. However, outside the specialized healthcare system, the knowledge of increasing challenges among PBTS often is not widespread, and consequently, adequate responses are lacking [50–53]. In this regard, there is evidence of the decline of available support to this population in aftercare, particularly when PBTS transition to the adult sector, highlighting the need for long-term follow-ups as well as clear information, both for survivors and providers, on available services and how to access them [21]. Coherent with the existing body of research, the responses obtained in the present survey underline that it is essential to sensitize the various sectors involved with PBTS to the specificities of this population. Another observation is that knowledge about existing support modalities directed to PBTS should be more widespread: cognitive training, intervention on health-related behaviors, assessment of neurocognitive and academic difficulties, career and vocational counseling, academic adaptations, and pharmacological and psychological interventions, to name a few [3,54]. As endorsed by the present sample, these interventions could be promoted by offering regular counseling with a healthcare professional or a specially trained stakeholder outside the clinic (Figure 4). They could also provide a better liaison and quick information transfer service from the pediatric to the adult sector, or from the healthcare sector to the education and labor environments. In this regard, in Europe, the PanCareFollowUp Recommendations Working Group developed the digital Survivorship Passport, a tool to facilitate the process of creating a personal care plan and sharing it with survivors' health professionals. This tool has been proven useful to emphasize awareness among survivors and health care providers in addition to tailored clinical evaluations and/or surveillance tests [55].

Respondents rated the solution regarding a possible support in relationships and sexuality as least helpful and their responses were more spread out, showing less agreement compared to the five solutions which received the most interest (CV = 52%). Interestingly, this solution was proposed by clinicians in the previous qualitative study on the basis of young survivors' repeated complaints regarding the lack of social relationships. It is

probable that professionals or the healthcare system might not be in the best position to respond to this specific need. This finding also underlines that institutional responses to meet population needs should always be validated with the target population, as surprises often emerge! In this example, the needs and expectations of PBTS may significantly differ from that of their clinicians, and, consequently, the type of response proposed by clinical teams may be inadequate [17,18]. Future research should be conducted to uncover ways to support PBTS dealing with isolation and loneliness.

We should recognize the limitations of this study. First, the sample is limited in number, and mostly concentrated in one province of Canada. Second, even though we tried to adapt the survey to the possible cognitive difficulties encountered by PBTS (e.g., concentration problems, learning difficulties), we cannot exclude that survivors who presented more severe sequelae preferred not to respond (or stopped, as some did early in the process), and as such, the sample may suffer significant selection biases which limit external validity. Moreover, because of the high level of potential participants (38%) discarded from the study, we hypothesize that the letter of presentation and initial communications, especially via social media, were too broad and caused several non-eligible people to access the online survey (i.e., older age, under treatment). Instead, we could have been more specific on our inclusion criteria. Third, we used four-point Likert scales to evaluate endorsements of proposed solutions. Alternative approaches could have yielded clearer differences, such as preference-based methods, or 7- or 10-point Likert scales, that would have enabled more nuanced endorsements to be expressed. Finally, even though we considered survivors and parents as independent samples, we cannot rule out that some survivors and parents may be from the same family units. As such, the results should be considered as exploratory. However, this study collected rich sociodemographic information and in-depth information about this vulnerable and understudied population focusing on a crucial period of survivorship, i.e., the return to daily activities. Candidate solutions to improve services for PBTS were designed and validated with end-users, which was an asset since it could facilitate future implementation processes.

**5. Conclusions**

Using an online mixed method survey among 68 respondents (43 survivors and 25 parents), we found sociodemographic specificities related to our sample of pediatric brain tumor survivors. We identified barriers and facilitators to their (re)entry to school and work in aftercare and based on their ratings, we are in a position to prioritize key solutions to improve re-entry in daily life activities: the need for regular evaluation, counseling and follow-up (Figure 4). Importantly, the present sample reported facing high challenges in aftercare and put forward a clear need for a tighter liaison of the pediatric healthcare environment with other sectors: the education and labor environments, as well as the adult healthcare sector. Future studies should evaluate the implementation of a better liaison among these different sectors and include the recent recommendations regarding the standards of care for the population of young cancer survivors [56].

**Supplementary Materials:** The following supporting information can be downloaded at: https://www.mdpi.com/article/10.3390/curroncol30090623/s1, Table S1: Long-form questionnaire—Young people version; Table S2: Experiences and difficulties of survivors in aftercare in health, school and work's domain; Table S3: Bonferroni post hoc test.

**Author Contributions:** Conceptualization and methodology, M.B., C.J.B., A.L.-J., É.D., M.L., M.-C.B., C.P. and S.S.; data analysis, M.B., L.-A.R. and S.S.; investigation M.B.; visualization L.-A.R. and A.L.; resources, É.R.; writing—original draft preparation, M.B. and S.S.; writing—review and editing, C.J.B., L.-A.R., A.L., A.L.-J., É.R., É.D., M.L., M.-C.B., C.P. and L.D.; supervision L.D.; funding acquisition, S.S. All authors have read and agreed to the published version of the manuscript.

**Funding:** This research was supported by the Sainte-Justine University Health Center Foundation, through endowments to the Center of Psycho-Oncology (CPO, Serge Sultan).

**Institutional Review Board Statement:** The study was conducted according to the guidelines of the Declaration of Helsinki, and approved by the Ethics Committee of UHC Sainte-Justine (#2021-2712).

**Informed Consent Statement:** Informed consent was obtained from all subjects involved in the study.

**Data Availability Statement:** The data presented in this study are available on request from the corresponding author.

**Acknowledgments:** The authors gratefully acknowledge all the young people, parents and clinicians who participated to the survey by sharing their experiences and helping us in this research's development. We also would like to acknowledge the Brain Tumour Foundation of Canada and Leucan association for their help in recruitment and their participation in the different steps of our study.

**Conflicts of Interest:** The authors declare no conflict of interest.

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
