# Peer review of "Prioritizing Solutions and Improving Resources among Young Pediatric Brain Tumor Survivors: Results of an Online Survey"

_curroncol, doi:10.3390/curroncol30090623_

Round 1
Reviewer 1 Report
The Authors present a very interesting paper: " Prioritizing solutions and improving resources among young pediatric brain tumor survivors: results of an online survey" because they managed a current problem from a social point of view related to long term survivors.
They had a good approach as, Aim, Introduction, Material and Methods, Results and Discussion as well as tables and figures . Anyway I would like to focus and have back answers from Authors of few points.
1. 38% of potential participants were discarded because they didn't meet the inclusion criteria: it is a quite large amount among a total of 109 potential participants: did the Authors ask to themselves why? letter of presentation of the study not enough good? did they made an effort to contact the potential participants in a different way (e.g. phone call?...)?
2.In the Discussion I didn't see any mention to the so-called "Passport" for cured subject promoted by the European Pancare Study group: uninteresting tool to follow the long term survivors
3. In the Conclusion it should be more readable to have a flow chart reporting the main steps to follow PBTS at the best. Especially to improve the social involvement of PBTS.
4. In the Discussion I would like to have a comment about the "oncological oblivion" that should be at the basis for a normal and good social reintegration.
Author Response
Thank you for all of your comments. They helped us improve the quality of our manuscript.
1) In regard to your first comment, we added a paragraph in the Limitations section (lines 478-482), where we hypothesized that our letter of presentation and the initial communications, especially via social media, were too broad which caused several non-eligible people to access the online survey (i.e. older age, under treatment).
2) In line with this comment, we added a few sentences (lines 450-454) to explain this useful tool that has been developed in Europe for cancer survivors and, thanks to your comment, we suggested its use as an interesting solution to emphasize awareness among survivors and health care providers in addition to tailored clinical evaluations and/or surveillance tests.
3) As suggested, we created a figure to report key solutions to help follow PBTS (lines 455-458).
4) Thank you for reminding us of this important issue. In this regard, we added a paragraph (lines 407-414) in which we highlighted the challenges related to financial discrimination towards cancer survivors and we pointed out studies which support the need to provide a legal framework based on the “right to be forgotten”.
Reviewer 2 Report
Interesting, important and strong study as the paucity of research with these survivors is certainly evident.
you did bring up significant information that can be disseminated and strongly encouraged to be done
You identified the need to adapt resources and investigate further some that got different perspectives i.e. from the survivors and the healthcare providers
the aftercare is certainly a major issue for these survivors of any childhood cancers.
you also recruit those up t 39 years and it is certainly important
on a final note the discussion is certainly thought provoking

Author Response
We thank you for your comments on our manuscript. According to comments coming from other reviewers we also improved the Discussion and Conclusion sections. Specifically, we clarified some limitations of the study and added a figure that outlines recommendations for young cancer survivors from our participants' findings and from studies regarding standards of care in this domain.
Reviewer 3 Report
Thanks for recommending me as a reviewer.In this paper, authors used a mixed-methods survey with 68 participants (43 survivors, 25 parents, PBTS’ age: 15-39 years). Firstly, we collected information about health condition, and school/work experience in aftercare. Then, authors asked participants to prioritize the previously identified solutions using Likert scales and open-ended questions. Authors used descriptive and inferential statistics to analyze data, and qualitative information to support participants’ responses. Participants prioritized the need for evaluation, counseling, and follow-up by health professionals to better understand their post-treatment needs, obtain help to access adapted services, and receive information about resources at school/work. If the authors complete minor revisions, the quality of the study will be further improved.
1. The introduction section is well written. However, it does not seem necessary to divide the background and purpose of the study into subsections.
2. The study method was adequately described.
3. Authors should add further limitations of the study in the discussion section.
Minor editing of English language required.
Author Response
Thank you for your comment.
1) As suggested, we unified the introduction section by putting together the background and purpose. We renumbered the manuscript sections consequently.
2) Thank you.
3) In line with this suggestion, in the Limitations paragraph (lines 478 - 482), we described the issues with the letter of presentation and the initial communications related to the recruitment step.